# A Systematic Review of the Use of Intraoral Scanning for Human Identification Based on Palatal Morphology

**DOI:** 10.3390/diagnostics14050531

**Published:** 2024-03-01

**Authors:** Sanjana Santhosh Kumar, Rachel Chacko, Amritpreet Kaur, Gasser Ibrahim, Dongxia Ye

**Affiliations:** 1Eastman Institute for Oral Health, University of Rochester Medical Center, Rochester, NY 14620, USA; sanjana_santhoshkumar@urmc.rochester.edu (S.S.K.); amritpreet_kaur@urmc.rochester.edu (A.K.); gasser_ibrahim@urmc.rochester.edu (G.I.); 2Department of Health Promotion and Behavioral Sciences, University of Texas School of Public Health, Houston, TX 77030, USA; rachel.chacko@uth.tmc.edu

**Keywords:** intraoral scanning, palate, rugae, forensics, human identification

## Abstract

A common application for intraoral scanners is the digitization of the morphology of teeth and palatal rugae. Palatal scans are most commonly required to fabricate complete dentures and immediate transitional dentures and serve as a reference point for assessing orthodontic results. However, they are also frequently included by accident, even though the main purpose of intraoral scanning is to reconstruct dentition using computer-aided manufacturing (CAM). The literature shows that the identification of disaster victims has frequently involved palatal rugae impressions. As the skull provides sound insulation, the rugae are resistant to heat, chemicals, and stress. Antemortem data might be difficult to find during a forensic inquiry, particularly in disaster victim identification cases. In contrast with DNA and fingerprints, there is a greater likelihood of having a dental record that contains palatal scans. With specialized software, the scans can be exported as open stereolithography (STL) files. Considering that a full case consumes up to about 100 MB of hard drive space, long-term storage should not be an issue compared to a plaster model. Additionally, dentists widely use online databases to exchange data for smile design, implant registration, and orthodontic purposes. This will produce a digital database that grows quickly and is readily usable for forensic investigations. The uniqueness of forensic features is frequently challenged; however, palatal morphology’s unique trait could make it possible as it is characteristic of individuals as well as the most distinguishing factor. This review will highlight how rugae, palatal morphology, mirroring, superimposition, and geometrics can serve in forensic identification.

## 1. Introduction

### 1.1. Background

In recent years, the collaboration of technology and dentistry has witnessed a paradigm shift with the advent of 3D intraoral scanners (IOS) [1]. This cutting-edge technology has not only revolutionized the field of dentistry but has also extended its applications to forensic odontology, specifically in the study of palatal morphology for human identification [2]. Forensic odontology is the branch of dentistry that deals with examining and evaluating dental evidence to be presented in the interest of justice [3,4]. Dental evidence in human identification is one of the main objectives of forensic odontology [5]. Due to its individualistic features, the palate is a complex and distinctive unique biometric marker. Understanding the intricacies of palatal morphology holds tremendous potential in the realm of forensic odontology, where accurate and reliable methods for human identification are important [6].

The palatal rugae are transverse ridges located in the anterior portion of the hard palate that constitute distinct anatomical features [7]. Scientific studies on palatal rugae patterns are referred to as palatal rugoscopy, and they have recorded that palatal rugae are highly consistent in shape throughout aging and are unique to individuals [8]. This inherent uniqueness underscores the significance of palatal rugae in forensic odontology. Numerous studies have documented the stability of palatal rugae, thereby establishing their potential as a valuable tool for forensic identification [9,10,11]. Furthermore, evidence suggests that the distinctiveness of the palatal rugae aids in racial profiling as their patterns exhibit specificity to racial groups. Additionally, they contribute to sex identification, enhancing their role in forensic investigations [12]. The palatal rugae remains the most extensively studied anatomical structure of the palate for human identification.

Besides the palatal rugae, the palate consists of important landmarks, including the incisive canal, the greater palatine foramen, and the lesser palatine foramen [13,14]. The domain of dentition also plays a predominant role in identification purposes, as teeth are resistant to extreme temperatures and decomposition [15].

Recent advancements in the use of IOS have gained much relevance in recent years, and they aid in capturing the entire dental arch, thus simplifying palatal scans. Intraoral scanners are 3D measuring medical devices capable of holistically capturing the dental arch, thereby producing 3D models of the entire dental arch, including the hard and soft tissue of the oral cavity [16]. Intraoral scanners are designed primarily to digitize dental structures and incidentally capture palatal morphology during routine clinical examinations. However, they are not customarily integrated into the planned examination routine for chief complaints. Studies have shown that the geometrics of the palate have some potential to provide individual characteristics of the human palate, thereby showcasing its usefulness as a screening tool in the human identification process [17,18]. Evidence suggests that IOS are considered reliable in the analysis of dental arch length and have been proven valid in assessing precision in invivo studies [19]. Morphological distinctions within the hard palate hold significance in the fields of forensic medicine, anthropology, anatomy, and scientific disciplines dedicated to exploring evolutionary development and population variations. Finally, IOSs could confirm, identify, and rule out false-positive situations by comparing the morphology of two palatal scans superimposed on one another [17].

As we delve into the existing body of literature, critical evaluation of available research will provide a comprehensive overview of the current state of knowledge, identify gaps in the literature, and suggest possibilities for future research in this upcoming field. As forensic odontology continues to evolve, the utilization of 3D intraoral scanners stands as a promising tool for advancing our understanding of palatal morphology and its significance in the process of human identification.

### 1.2. Objectives

This systematic review aims to address the technical capabilities of 3D intraoral scanners, their potential applications in forensic odontology, and their accuracy in capturing palatal morphological data in establishing individual identity.

## 2. Methods

### 2.1. Protocol and Registration

The Preferred Reporting Items for Systematic Reviews and Meta-Analysis (PRISMA) guidelines were followed in the conduct of this systematic review [20], and it was submitted to PROSPERO and registered under the ID CRD4202451406. The checklist of the PRISMA guidelines is presented in Appendix A.

### 2.2. Search Process

The search was conducted between December 2023 and January 2024 in the databases PubMed, Embase, Web of Science, Dentistry and Oral Sciences, and Google Scholar. A set of keywords and MeSH/Emtree terms about intraoral scans, palate, and forensics were incorporated into the search. The Boolean operators AND and OR were utilized to combine the search phrases. The search procedure was modified slightly to fit each database. Each database’s specific search methodology is described in Appendix A.

### 2.3. Eligibility Criteria

In this systematic review, studies with primary data on intraoral scanning of the palate evaluated for human identity or forensics were considered, with no restrictions on published language, with a longitudinal or cross-sectional design.

If the articles did not measure palatal changes, utilized cast models, evaluated scans from cadavers, or used scanners other than an intraoral scanner, they were eliminated. Exclusion criteria also applied to case reports or case series, conference papers, letters to the editor, reviews, meta-analyses, opinion pieces, and study procedures that did not present original data.

### 2.4. Study Selection

After the search was completed, duplicates were eliminated, and relevant articles were identified from the titles and abstracts. The inclusion and exclusion criteria suggested for this systematic review were taken into consideration while reading the entire text to ascertain eligibility. Two independent researchers (S.S. and R.C.) carried out the systematic search, study selection, and data extraction. A third researcher (D.Y.) addressed any uncertainties or differences.

### 2.5. Data Extraction

Information about the authors of the studies, the sample characteristics, the number of subjects, the type of intraoral scanner used, the time points of scanning, the area of the palate scanned, the type of software used, comparison groups, and the results of statistical analyses were extracted for each included study.

### 2.6. Quality Assessment

Two reviewers, S.S. and R.C., evaluated the quality of the included papers using the Critical Appraisal Skills Programme (CASP)—cohort study checklist, a method designed to evaluate the possibility of bias in the outcomes of cohort studies that report on the accuracy of palatal morphological stability for use in forensics. The outcomes of the included studies were assessed using twelve points divided into three main categories—are the results of the study valid? (Section A); what are the results? (Section B); and will the results help locally? (Section C). Whenever there was a disagreement, D.Y., the third reviewer, was consulted until consensus was reached.

## 3. Results

### 3.1. Literature Search

Our preliminary search yielded 42 records in total. Following the elimination of duplicates, 28 studies remained. After titles and abstracts were screened, seven articles were disqualified. A total of 14 full-text articles were eliminated because they failed to meet the inclusion or exclusion criteria. Finally, seven studies were included in the current systematic review. The detailed search process and exclusion criteria can be found in Figure 1.

### 3.2. Quality Assessment

The methodological quality of the included studies is summarized in Appendix A. Although only three studies evaluated scans at different time points, all included studies measured the outcomes accurately, thus emphasizing the reliability of 3D palatal scans for human identification.

### 3.3. Study Characteristics

In all included studies, the number of participants ranged between 3 to 199 [17,21,22,23,24,25,26]. Only two studies reported the gender of the participants [23,24] and one reported the type of scanning pattern they followed [17]. Since the scanning pattern has a major impact on the speed and precision of digital impressions [27], Yang et al. suggest that the palatal side of the posterior teeth is where the initial scan should start for palatal scanning. The full palatal scan should be completed together with the arch [28]. The sample type was not reported in two of the seven included studies. Emerald, iTero, and Sirona IOSs were majorly used in all included studies [24,25]. Three studies [21,24,25] included the time points of when scans were taken, whereas all seven studies reported the type of software they used to store the scans [17,21,22,23,24,25,26] (Table 1).

### 3.4. Synthesis of the Results

#### 3.4.1. Outcome of Comparison Groups in the Included Articles

Comparison between direct and indirect scans of the palate was carried out in two studies [21,25], whereas the other five studies compared palatal morphology, dentition, and mirrored and superimposed scans [17,22,23,24,26]. The deviation in the morphology of the palate was noted to be high in indirect scans, truncating its reproducibility compared to IOSs [25]. The same study also found that the same scan produced from different types of scanners could also lead to reduced repeatability of palatal morphology. Simon et al. in three of his studies reported that intra scans of twins had significantly fewer differences compared to inter scans between them [17,23,26]. Detailed results of the comparisons can be found in Table 2.

#### 3.4.2. Main Study Outcomes of the Included Articles

All seven included articles reported no significant intra-scan changes or deviations. Scans of different participants were compared with each other in five studies [21,22,23,24,25,26]. All of the articles reported no statistically significant mean difference between scans of the same participants at different time points, suggesting that 3D digital models from IOSs are a highly valuable tool for human identification. On the other hand, six articles highlight the distinctiveness of the palate in forensics, by showing a significantly higher variance in palatal morphology when scans of various people were compared with one another [21,22,23,24,25,26]. Table 3 discusses the main study outcomes in detail.

## 4. Discussion

To facilitate fast and precise identification, the discipline of forensic odontology is now using digital images of teeth; IOSs will raise the value of using digital images [29]. In this study, we broadly chose studies that emphasized the use of intraoral palatal scans for human identification purposes [17,21,22,23,24,25,26]. These studies describe the significance and stability of palatal rugae, and the geometry, mirroring, and specific 3D landmarks of the palate for their use in forensics. In two of the seven articles, orthodontic therapy did not affect the palate’s morphology [21,25]. Several studies also confirm this factor [2,30,31,32]. There is controversy in the change in position of the incisive papilla in orthodontically treated patients to use it as a reliable landmark for forensics [21,33,34,35].

Taneva et al. were the first to utilize a three-dimensional way of evaluating the rugae pattern to identify individuals [21]. Using digitalized alginate imprints, plaster models, and direct intraoral scans, they examined the 2D (measured on sections) and 3D deviation of the medial and lateral extremities of the rugae. Their research indicates that compared to the 2D method, the 3D method is more trustworthy for human identification [21].

Previous literature has demonstrated that palatal dimensions, arch depth, morphology, incisive papilla size, and palatal rugae could be indicative of sexual dimorphism in serving as sex predictors (gender-specific) [36,37,38]. In Negishi et al.’s study, a moderate to high genetic contribution was found to influence palatal morphology, with the posterior palate having a greater genetic contribution to height than the anterior [39]. Boton et al., showed that palatal geometry evaluated on an intraoral scan may identify victims and people with moderate certainty, even allowing for the distinction of identical twins and strangers [17]. Once potential matches have been chosen, their identity can be verified with one more palatal scan superimposition [22,30]. If there is low variation between two scans, it could be quite useful to make a mirrored alignment between them in a forensic inquiry [26]. In addition to helping locate the victim’s family, the mirroring technique can verify a potential twinning relationship [26]. Four of the seven included studies in this review reported no statistically significant deviation in palatal morphology between scans of siblings, making the palate more individual-specific [17,22,23,26].

A previous study suggested that in palatal side-to-side asymmetry, there may be a relatively small (0.3–0.4 mm) difference in their sizes between the left and right sides measured in the horizontal plane [30]. The major limitation of dental records could be that dentists may sometimes focus on quadrant rehabilitation, limiting the scan to only one palate side. In cases of severe accidents, half of the victim’s palate could become damaged. Simon et al. suggest that, if the contralateral side is available in the antemortem or postmortem database, the degree of the mismatch between mirrored and non-mirrored counterparts could be huge. This would make the accuracy of the palatal scan-based identification method possibly challenging or impossible [26]. In another study, the same author noted an interesting fact that despite having nearly identical DNA, the MZ twins’ palate morphology reveals discrepancies, suggesting that the reproducibility of palatal intra-oral scanning could be helpful in forensic odontology [23].

### 4.1. Palatal Scans Using Different IOSs

Different types of intraoral scanners are available, like the iTERO Element 5D plus, CEREC Primescan, Medit 1700, Omnicam, Dexis IS 3700, Trios 3, and 3Shape Trios 5 Wireless [40,41,42,43]. Overall, the highest precision outcomes were seen with iTero5D, Medit i700, and Omnicam [40]. In another study, the authors reported that deviations in trueness and precision between these scanners would not be clinically relevant since the recorded accuracy values were of clinical acceptability. The iTERO Element 5D plus is the fifth generation of the series. It is known to be incredibly fast in recording the entire arch within 30 s [44]. Its impressive software capabilities are that it can simultaneously record 3D, intraoral color, and near-infra-red images (NIRI). The NIRI serves as a caries-detecting technology. One of its drawbacks can be the size of the wand, which is large and can affect maneuverability and comfort. The CEREC Primescan has similar characteristics but could be a bit costlier [44]. On the other hand, the Medit 1700 is an affordable scanner while also being user-friendly. However, its drawback is that it lacks CAD/CAM software and does not facilitate the purpose of same-day dentistry as it requires third-party mills and software [45]. It does not have caries-detecting software nor are its scans accepted for Invisalign purposes. The Trios 5 is a recent introduction to the market, with modifications made to its size for better comfort and maneuverability, and it operates wirelessly [1]. Comparative studies of these scanners have evaluated the precision of these scanners and have discussed the fact that the TRIOS 3 showed higher precision in regard to the recording of the palate and dental arch [43]. In another study, several IOSs were used to scan the maxilla of an edentulous cadaver: iTero Element 2 (Align technologies), Cerec Primescan (Densply Sirona), Trios 4 (3Shape), Trios 3 (3Shape), and Medit i500 (Medit). Out of all of the scanners, the palatal scan taken using Medit i500 exhibited a significantly lower trueness. However, the authors mentioned that the amplitude of trueness and precision deviations from all scanners were still below the threshold level, therefore rendering no clinical significance [46]. Also, between CS3600 and TRIOS3, palatal area scans when superimposed showed no significant difference [43].

### 4.2. Influence of Scanning Technique

The impact of the operator’s experience on scan accuracy and operating time indicates that shorter scanning times and the operator’s expertise increase the accuracy of a digital scan [47,48,49]. There is abundant literature on the scanning technique for edentulous arches [50,51,52,53]. Since edentulous arch impressions are mainly used for the fabrication of complete dentures, these involve scanning of the palate in its entirety; studies that measured the accuracy of the scanning technique of intraoral maxillary edentulous scans were reviewed for discussion. Zarone et al. scanned typodonts with and without rugae using three different scanning techniques: buccopalatal, palatobuccal, and s-shaped technique. They discovered that the mean values for trueness and precision for the buccopalatal technique—moving counterclockwise along the palatal vault and then longitudinally in the posteroanterior direction—displayed greater mean values for trueness and precision than the palatobuccal technique [52]. Additionally, it has been reported that an S-shaped scan, which begins at the left maxillary tuberosity and moves the scanner tip in an S-shaped pattern along the ridge to alternate between the palatobuccal and buccopalatal areas, is advised to minimize errors. This ensures that there is sufficient overlap with previously scanned areas [50]. For our study, in Figure 2, we used the “S”-shaped scanning technique to cover the dentition and zig-zag pattern [23] to scan the palate of our volunteer. Jung et al. evaluated the IOS accuracy of the maxillary and mandibular arches’ supporting tissues. The authors proposed that using a scanner with specially designed tips for soft tissue targeting could yield better outcomes [54].

### 4.3. Stability of the Palatal Area in Forensics

The palate has been observed to exhibit resilience against adverse thermal conditions in survivors of burn injuries. Studies and observations indicate that the palate shows significant resistance to heat-related damage [55,56]. The stability and significance of palatal morphology in forensic odontology have been well-documented in various studies. Comparisons of palatal rugae with fingerprints is a widely accepted method in human identification, further emphasizing the robustness of palatal morphology in this context [55,57]. Another relevant finding establishes the suitability of palatal rugae as reference landmarks for longitudinal study [58]. They have not shown significant changes during short observation studies involving cast analysis of transverse and anterior–posterior planes [58]. Moreover, advancements in technology such as the use of intraoral scanners have expanded the scope of studying palatal rugae, thereby enhancing the efficiency and accuracy of human identification processes. Intraoral 3D scans of the palate have proven to establish good morphological reproducibility rates when compared to cast models, even though in real-life scenarios the palate could be subject to change. However, since these changes are not significant statistically, palatal morphology is considered important in forensic odontology [25]. The palate, however, is subjected to change post orthodontic treatment. Although these changes lack statistical significance, there are documented instances of alterations that could affect stability. However, the third rugae is suggested as a stable and reliable reference point even in post-orthodontic treatment cases [59,60]. Another study compares maxillary dimensional alterations in adults with a cleft lip and palate at two treatment phases: one after orthodontic treatment and the other after implant-supported prosthesis delivery. The comparison establishes dimensional changes in the arch post-treatment [61]. Hence, the palate serves as a reliable and stable reference point in forensic science, with some considerations in palatal expansion cases [60].

### 4.4. Reliability of IOSs Compared to Dental Cast Models

Several studies have investigated the accuracy of intraoral scans, which encompasses two parameters, namely ‘trueness’ and ‘precision [62,63,64]. Intraoral scanners can create an imprint of the oral cavity that is as near to its original shape as possible, free from distortion or deformation [43]. Prior research examined trueness and precision using models created by stone casts and traditional impressions [28,65]. When comparing dental cast models and IOSs, overall, there was only a slightly higher discrepancy in the cast models, making it on par with IOSs [66]. This is possible mostly when custom trays instead of stock trays are used. Nevertheless, this may not be ideal, particularly in the case of palatal structures, where the original anatomy may be altered by soft-tissue surface pressure points created during the traditional impressions [25,43]. When comparing conventional impressions with intraoral scanning of a cleft palate, Okazaki et al. discovered that a 3D printer model was safe because there was more tissue displaced in the conventional impression due to pressure applied during impression taking [67]. Table 4 below highlights the difference between palatal scans of direct (IOS) and indirect (dental cast) models.

Physical impressions and dental cast models have poor technical reproducibility since bubbles or distortions are common in them. Since investigation time is limited in human cadavers, repeated physical impressions might damage the decayed tissues. An IOS also eliminates the need for the disinfection and cleaning of dental impressions and trays and bypasses the traditional process of casting models, preventing wear on the model and facilitating swift communication and accessibility [41]. Additionally, with shorter working time, digital impressions are more comfortable and widely accepted by patients than conventional impressions [71]. Three-dimensional technology, due to its superior nature, has overcome 2D analyses in several ways. A recent study noted that even 3D extraoral scanners are still far better than conventional dental models in terms of having higher precision and trueness [72].

### 4.5. The Influence of Dental Treatments on the Morphology of the Palate

Dental treatments, as well as oral diseases, could significantly affect the morphology of the palate [73,74]. Based on Martins dos Santos’s palatal rugae classification, the most prevalent palatal rugae shape in aggressive periodontitis was found to be angle followed by sinuous, and in chronic and aggressive periodontitis, it was found to be sinuous followed by line pattern. The differences in rugae shape between the three groups may be attributed to genetic factors, disease progression, and recent shared ancestry, which has probably rendered their differences to moderate levels [74].

#### 4.5.1. Orthodontics

The non-invasive nature of intraoral scanners, combined with their ability to scan the entire arch, including the soft tissues, makes it simple to analyze the mouth’s dimensions in orthodontics [17,23,75]. Measuring 3D palatal models could be a reliable way to assess symmetry, morphology, and orthodontic treatment outcomes [30]. Palatal rugae, in orthodontics, have been considered a standard for the superimposition of upper jaw models when artificial markers (such as mini screws) are not accessible [76,77]. Although palatal rugae are considered the most stable landmarks of the oral cavity [78], palatal expansions, which are often carried out to make more room for crowded teeth, significantly influence the morphology of palatal rugae [79,80]. In individuals who have previously had palatal expansions, human identification and serial superimpositions based on palatal rugae should be considered cautiously and not be carried out immediately after expansion [79,80]. Palatal rugae are mostly stable in shape and number, not position, after rapid maxillary expansion [32,81]. On the other hand, a study revealed that slow maxillary expansion had no discernible effect on the growing individuals’ palatal rugae pattern [11].

Pazera et al. assessed changes in the location of the palatal rugae relative to the underlying maxillary skeletal structures in 24 growing patients receiving fixed orthodontic treatment without palatal expansion, with a treatment period varying from 1.5 to 3.5 years. Based on the maxillary structural superimposition, the papilla point and all rugae points’ median anteroposterior positions remained constant and showed no statistically significant variations [31]. While the medial and lateral locations of the third palatal rugae are stable landmarks for assessing tooth movement, the stability of the first and second palatal rugae depends on the kind of orthodontic tooth movement [82]. Therefore, if palatal rugae are regarded as an individual landmark, we can refer patients for scanning of their palatal rugae following orthodontic treatment to record their post-treatment pattern. It should be emphasized that treatment relapse may alter the post-treatment pattern again [81].

Palatal symmetry is an environmental phenomenon, and IOSs helped confirm that orthodontics is generally unaffected as they are relatively symmetrical in most human beings [30]. The literature suggests that the results of rapid maxillary expansion provide more parallel palatal expansion during the early treatment phase as opposed to a V-shaped opening during the latter treatment phase [83,84]. Guidice et al., in their study about palatal changes after correction of posterior cross-bite, reported that IOSs confirmed slight morphological asymmetry of the maxillary anatomy of the palate mainly confined to the lower part of the palate at the level of alveolar processes [85].

#### 4.5.2. Surgery

Dental extractions of the maxillary arch often affect the palatal form. According to a study, palatal length decreased 61% more in the after-extraction group compared to the non-extraction group [86]. In another study, a decrease in palatal volume was noticed in extraction patients [87]. As far as rugae are concerned, significant alterations were seen in the rugae location, particularly at their lateral ends, corresponding to the direction of tooth migration following the loss of neighboring teeth and the bone resorption around the maxillary arch [73]. However, variation mainly exists in the first rugae in cases of extraction, thereby making palatal rugae still distinct, nevertheless, and they can be an additional resource for individual identification [88]. When orthodontic treatment involves extraction, there is a possibility of changes in the length, width, and depth of the palate compared to orthodontics without extraction, which changes only the depth of the palate [89]. Kratzsch and Opitz reported that after surgical repair of a cleft palate, the rugae counts per segment reduced significantly; however, the third rugae was never lost following surgery [90]. In the case of a cleft lip, Kramer et al. reported that the maxillary arch depth reduced after surgery and was compensated by continued anteroposterior palatal growth [91]. Past research indicates that cleft palate surgery and forced canine eruption limit the palatal rugae’s usefulness as a forensic marker [57,90,92]. According to Camargo et al., rugae regions should not be chosen for the palate donor site in gingival graft surgery as they may persist in the grafted tissues [93].

#### 4.5.3. Prosthetics

There is not much evidence about the impact of dental prosthetics on the morphology of the palate. Instead, palatal prints from partial and complete dentures have been used as valuable tools in identifying victims [10,94]. However, following implant therapy in the alveolar cleft region, the dental arch may undergo a slight reduction in width, altering its geometry [95]. Prolonged use of complete dentures imparts pressure and compression on palatal mucosa, causing histological changes in palatal rugae [96,97]. This in turn changes the length of rugae but does not affect their number and orientation [36,98].

### 4.6. Limitations of the Available Literature

The main limitation of the available literature is the evaluation of the reliability and reproducibility of intraoral palatal scans in a smaller sample size and over a shorter time point. Gender and ethnicity were not considered in any studies. In total, six out of the seven studies chose only the young population group as their sample [17,21,22,24,25,26]. Additionally, five out of the seven included studies did not report the scanning technique they used [21,22,24,25,26] as it plays a huge role in the precision of the digital impression [27]. Only two studies evaluated participants who were previously treated with orthodontic appliances. None of the studies included participants who had deformities of the palate (soft and hard tissue) or any history of palatal surgeries, thus limiting the evidence.

### 4.7. Future Research Direction

It will be interesting to assess the morphological changes of the palate and palatal rugae landmarks in each decade of life to determine whether the aging and growing of the palatal mucosa result in any notable alterations. Data that includes people with a wide-ranging medical history and previous history of palatal surgeries and palate anomalies could be useful. According to Camargo et al., rugae areas should be avoided when choosing a palatal donor location for gingival transplant surgery since they may persist in the grafted tissues [93]. For this reason, palatal soft tissue may or may not be useful in forensics; research on the impact of palatal tissue harvesting for regenerative surgery could highlight this. According to earlier research, width gives the highest level of ethnic discrimination [99,100]. To confirm the use of automated geometric measurement as an ethnicity discriminator, further soft tissue scans of different ethnic populations will be required. Since maxillary arch depth can be utilized in conjunction with other morphometric techniques to determine a person’s sex [37] for a specific population, the mean value of the palatal and maxillary arch depths can serve as a baseline value and point of reference for future research. With the advancements in technology using artificial intelligence, the creation of a fully automated comparative procedure could also be possible in the future through which the identification process could be hastened. Original research of matching between antemortem and postmortem data obtained by new IOSs will provide increased evidence of the reliability of intraoral palatal scans in forensics. Furthermore, research should also focus on their application to deceased people, providing an additional primary identification feature, for example, in a mass disaster.

## 5. Conclusions

During this time of modern dental treatment, IOSs of the palate will contribute to the forensic applicability of dental data in human identification. Palatal uniqueness in its dimension, landmarks, and palatal rugae in 3D digital palatal models could serve as a highly reliable tool for human identification, especially during road traffic accidents or mass disasters as the palatal area is less likely to become damaged. This also stands true for edentulous victims as there would be no dental component for identification. After dental work is finished, it is crucial that dentists save their digital models. In the present and the long-term future, these preserved models will be of great assistance for forensic, legal, and rehabilitative purposes.

## Figures and Tables

**Figure 1 diagnostics-14-00531-f001:**
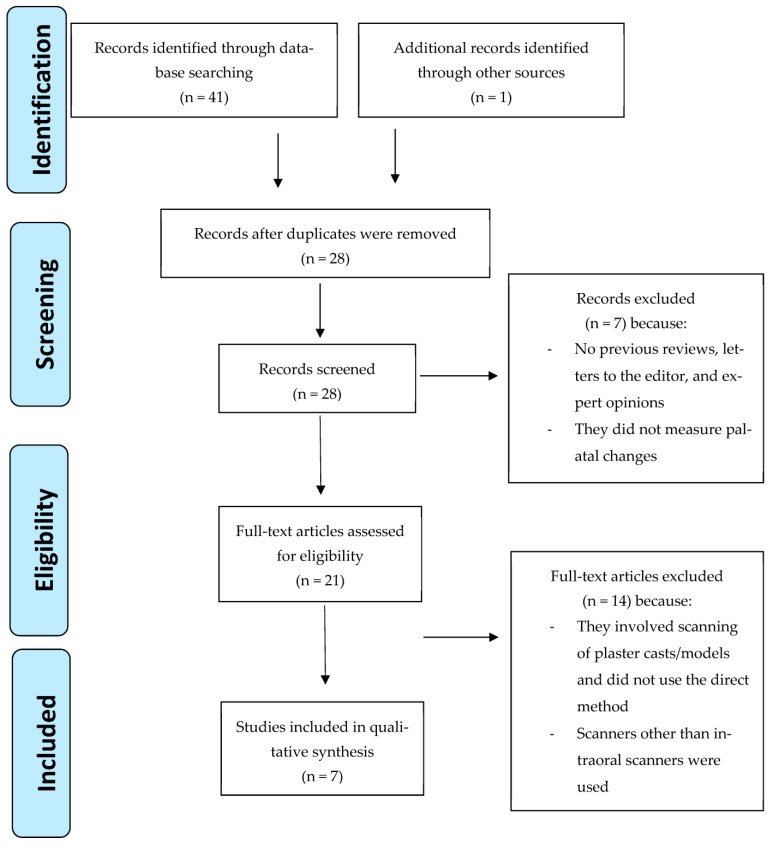
A flow diagram of the search strategy conducted (PRISMA flow of study selection process).

**Figure 2 diagnostics-14-00531-f002:**
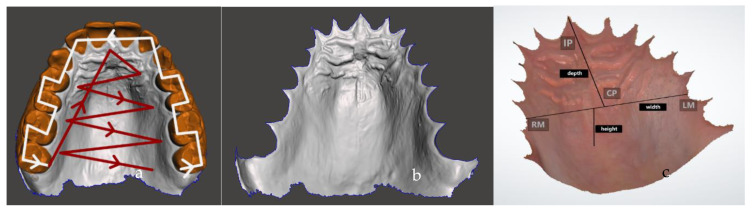
The representative intraoral scan was acquired from a volunteer subject with an iTero scanner, started by a “S” movement on dentition (white arrow) and a zig-zag movement from the incisive papilla and finished at the border of the hard and soft palate (red arrow) (**a**). Teeth were removed from the scan, and only the palatal area was kept; the STL file was generated (**b**). The most common 2D and 3D reference landmarks were demonstrated, such as RM: a point at the gingiva of the right first molar, LM: a point at the gingiva of the left first molar, IP: the most anterior point of the incisive papilla, CP: the central point, width: the distance between the RM and LM, and depth: the distance between the IP and CP (**c**).

**Table 1 diagnostics-14-00531-t001:** General characteristics of the included articles.

Literature	Type of Study	Sample Size	Age (Years)/Gender M:F	Sample Type	Type of IOS	Time Points of Scan	Scanning Technique	Type of File/Software Used to Export	Statistical Analysis
Taneva et al. (2015) [21]	Pilot study	20	12–18 years/NA	Adolescents	OrthoCAD (Align Technology, Inc., San Jose, CA, USA), Ortho Insight 3D™ laser scanner (Motion View Software, LLC, Chattanooga, TN, USA), and iTero^®^ Intra Oral Digital Scanner (Align Technology, Inc., San Jose, CA, USA)	Initial scan—before ortho treatment; second scan—during ortho treatment; roughly 20–24 months apart.	NR	Stereolithography binary file format (*.stl)/Geomagic^®^ Software (Geomagic^®^, Research Triangle Park, NC, USA.	Descriptive and comparative statistics were performed using SPSS 20.0 (Chicago, IL, USA)
Simon et al. (2021) [22]	Pilot study	3	17, 22, 26 years/NA	Monozygotic twins	Emerald intraoral scanner (Planmeca, Helsinki, Finland, software version: Romexis 5.2.1)	NR	NR	NA/GOM Inspect software (GOM GmbH, Germany)	Generalized linear mixed method using SPSS (IBM SPSS Statistics for Windows, Version 27.0., USA)
Simon et al. (2022) [17]	Cohort study	176	NA	61 monozygotic twin pairs and 27 dizygotic twin pairs	Emerald^®^ intraoral scanner with ROMEXIS^®^ PlanCAD Easy software (version 5.2.1, Planmeca Oy, Helsinki, Finland).	NR	Standard scanning pattern	NA/GOM Inspect^®^ 3D mesh processing software (Suite 2020, GOM GmbH, Braunschweig, Germany, Meshmixer—(version 3.5, Autodesk Inc., San Rafael, CA, U.S.A.)	Mean absolute deviation (MAD), linear discriminant analysis (LDA), and Bayesian theorem
Simon et al. (2020) [23]	Cohort study	201	17–74 years/54M:147F	64 monozygotic twins, 33 same-sex dizygotic twins, and 7 opposite-sex dizygotic twins	Emerald^®^ intraoral scanner (Planmeca Oy, Helsinki, Finland, software version Romexis 5.2.1	Scanned thrice at the same time: R1, R2, and R3	Zig-zag scanning pattern (starting from the incisive papilla and finishing at the border of the hard and soft palate)	Stereolithography binary file format (*.stl)/GOM Inspect^®^ inspection software (GOM GmbH, Braunschweig, Germany	Generalized linear mixed model with gamma-distribution and log-link function
Bjelopavlovic et al. (2023) [24]	Longitudinal cohort study	105	19–38 years/37 M:68F	NR	Omnicam SIRONA ^®^	Initial scan; second scan: 3 months later	NR	Stereolithography binary file format (*.stl)/Cloud Compare (v. 2 12.0)	STATA 17 (STATACORP 2022, Revision 10)using a *t*-test
Mikolicz et al. (2023) [25]	Retrospective cohort study	40	18–32 years/NA	NR	Emerald intraoral scanner (software version 5.2.1, Planmeca, Helsinki, Finland) IOSs Emerald S (software version 5.1.3.7, Dentsply Sirona, Charlotte, North Carolina, USA)	Initial scan: 2019; second scan: 2021	NR	NA/GOM Inspect^®^ engineering analysis software (Suite 2020, GOM GmbH, Braunschweig, Germany	Mean absolute deviation (MAD) was evaluated using a generalized linear mixed model using gamma distribution with a log link function and the Kruskal–Wallis non-parametric test [SPSS version 28 (IBM)]
Simon et al. (2023) [26]	Cohort study	174	NR	61 monozygotic twin pairs and 26 dizygotic twin pairs	Planmeca Emerald (Planmeca Oy, Helsinki, Finland, version number Romexis 5.2.1	NR	NR	GOM Inspect Suite software (GOM GmbH, Braunschweig, Germany	Wilcoxon test

NR—not reported; NA—not applicable.

**Table 2 diagnostics-14-00531-t002:** Outcome of comparison groups in the included articles.

Literature	Area of the Palate Scanned	Covariate	Comparison Groups	Outcome of Comparison
Taneva et al. (2015) [21]	Posterior point of the IP, and the most medial and lateral endpoints (12 nos) of the palatal rugae	Orthodontic management (no significant difference)	Comparison between ortho insight 3D™ plaster model scans (Motion View Software, LLC, Chattanooga, TN, USA) and iTero^®^ intraoral scans (Align Technology, Inc., San Jose, CA, USA)	All 2D variables had statistically significant differences, indicating that 2D images and linear measurements are not useful for human verification
Simon et al. (2021) [22]	Palatal surface	Not present	Comparison of palatal deviation with teeth deviation between siblings	The palatal deviation between siblings was 3–4 times higher (0.393 ± 0.079 mm, *p* < 0.001) than the teeth deviation
Simon et al. (2022) [17]	Palatal width, height, and depth (occlusal first and S-shaped)	Not present	Comparison between original and smoothened scans	The intra-twin original scans were not statistically significant from smoothened scans in both M.Z.T. and D.Z.T., *p* = 0.06, and *p* = 0.28
Simon et al. (2020) [23]	Palatal surface superimposition	Not present	Comparison of intra-twin deviation between M.Z.T and D.Z.T.	Intra-twin deviation of monozygotic twins (406 ± 15 μm) was significantly lower than that of dizygotic twins (594 μm ± 53 μm) *p* < 0.01
Bjelopavlovic et al. (2023) [24]	Palatal rugae pairs	Not present	Comparison between inter- and intraindividual differences of palatal rugae pairs	The intraindividual differences were highly significantly lower than the interindividual differences (*p* < 0.0001)
Mikolicz et al. (2023) [25]	Anterior part of the palate	Orthodontic management (no significant difference)	Comparison between (a) different IOSs and (b) IOSs with physical impressions and stone casts	(a) The higher deviation of forensic reproducibility compared with technical reproducibility and repeatability was caused by differences in the scanners; (b) the forensic and technical reproducibility of IOSs was 38–40 µm and 3–4 times higher than the physical impression
Simon et al. (2023) [26]	Palatal surface mirroring	Orthodontic management (recorded but not measured)	Comparison between monozygotic and dizygotic original and mirrored scans	Significant difference between palatal surfaces of monozygotic and dizygotic original and mirrored scans, *p* < 0.001

IOS—intraoral scanner; M.Z.T—monozygotic twins; D.Z.T—dizygotic twins; 2D—two-dimensional; 3D—three-dimensional.

**Table 3 diagnostics-14-00531-t003:** Main Outcome of the included articles.

Literature	Interscan Changes	Intrascan Changes	Overall Outcome
Taneva et al. (2015) [21]	No significant mean difference between 13 3D landmarks; *p* > 0.05	No significant mean difference between 12 3D landmarks; *p* > 0.05, except for the posterior point of the IP *p* < 0.05	Three-dimensional landmarks help with the matching process; hence, 3D digital models are a highly effective tool in evaluating different palatal rugae patterns with accurate landmark identification
Simon et al. (2021) [22]	The mean absolute deviation of the palates of non-relatives was significantly higher (1.061 ± 0.314 mm, *p* < 0.001)	Palatal deviation was significantly lower than non-relatives (*p* < 0.001)	Palatal uniqueness in 3D digital palatal model could serve as a highly reliable tool for human identification
Simon et al. (2022) [17]	NR	Geometrical comparison of intrascans resulted in 91.2% sensitivity and 97.8% specificity	Three-dimensional data containing only palatal height, width, and depth without surface morphology could assist with human identification
Simon et al. (2020) [23]	Superimposition of scans of two siblings had a higher deviation value	Superimposition of two scans of the same subject had smaller deviation values	Monozygotic twin siblings are highly distinguishable from one another and may represent individuality for the entire human community
Bjelopavlovic et al. (2023) [24]	Scans of randomly matched pairs had larger variance and precision difference greater than 300 μm from the mean	All repeated scans of one participant had precision values less than 300 μm around the mean	Palatal fold pairs do not differ at different lifetime points and are highly individual specific and differ significantly from individual to individual
Mikolicz et al. (2023) [25]	The precision value of scans between siblings (239 μm) was much greater than the highest forensic reproducibility value (141 μm)	The anterior palatal area showed significantly better repeatability and forensic reproducibility than the whole palate (*p* < 0.001)	The anterior area of the palate is a good candidate for identification purpose, owing to its inclusion in antemortem scans and its good reproducibility
Simon et al. (2023) [26]	In 22–27% of the twins, the difference between scans decreased after mirroring, suggesting that these twins have a contralateral similarity	The mirroring of the replicated scans increased the surface difference between the original and mirrored one by 7–9 folds, suggesting a significant asymmetry of the palate	(a) The between-sibling values were much higher than the between-replicate ones, indicating that an intraoral scanner is reliable for distinguishing persons. (b) The discrepancy between mirrored and non-mirrored scans indicates that if the contralateral side is available, the accuracy of the palatal scan-based identification process could be challenging

NR—not reported; IP—incisive papilla; 3D—three-dimensional.

**Table 4 diagnostics-14-00531-t004:** Difference between palatal scans of direct (IOS) and indirect (dental cast) models.

Factors	Dental Cast	Intraoral Scan
Accuracy	Not adequate [68]	Highly accurate [68]
Validity and Reliability	Not adequate [21,69]	Highly valid and reliable [21,69]
Deviation of palatal trueness	Affected by the flexibility of palatal soft tissue [28]	Not affected by the flexibility of palatal soft tissue [28] but increases with an increase in arch width [65]
Reproducibility	Not adequate [25,70]	Good reproducibility [25,70]
Arch dimension (anteroposterior and transverse)	Same as an intraoral scanner [18]	Same as a dental cast [18]

## Data Availability

Not applicable.

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
