# Peer review of "A Systematic Review of the Use of Intraoral Scanning for Human Identification Based on Palatal Morphology"

_diagnostics, 2024, doi:10.3390/diagnostics14050531_

Round 1
Reviewer 1 Report
Comments and Suggestions for Authors
Thank you for your interesting work concerning human identification based on palatal morphology.
I have some remarks and suggestions which can certainly improve this interesting work.
You mentioned in the end 7 articles which could be included in this review. Nevertheless, the question is if the palatal rugae are a constant anatomical marker like a fingerprint which does not chance during lifetime. Regarding this, there is only one article which investigated children and young adults in the time span of 20-24 month. The studies out of the 3 which investigated 2 time points exanimated adult patients. So, the discussion and also conclusion should focus on the rugae being a constant feature and not mainly on the accuracy of intraoral scans.
So, the conclusion that the palatal rugae are a highly reliable tool for human identification is in my opinion a littelbit misleading, because evidence this missing.
I would also appreciate to include the methods in forensic odontology used for identification. Usually ones prefer to identify prothetics like inlays, crowns and bridges which are characteristic and it is not uncommon that general dentists are asked if the dental status of the human to be identified is fitting the status or if dentists remember characteristic restorations. Only in case these options are missing, the rugae might be a hint for identification.
Author Response
Hi,
Please find the attached document for response.
Thank you.

Reviewer 2 Report
Comments and Suggestions for Authors
Dear Authors,
very welcomed manuscript in the field. Clear aim, almost correct analysis and well developed responses for the all research questions.
However, I have also a few minor objections for some manuscript chapters:
1) please, indicate the time duration (from-to) for the literature search and name the data basis used for this purpose also in the text (not only in the Suppl. File 2);
2) please, indicate also the year for the Taneva et al., and the same for the last 3 mentioned sources in the Table1;
3) please, decipher the abbreviations after the all Tables;
4) summarize also the main findings of the Table 2 also in the text in way of one paragraph;
5) Excellent Discussion, I really liked the subdivisions very much. Thank you!
6) well, the Literature sources request more careful arrangement. Please, go through once more, add the missed info for the 27, 74, 95 and check the compliance with the requirements! Also think please do you really need these 7 previous century sources (out of 101) for you otherwise very nice manuscript. Do not fit for this scientific Journal!
Author Response
Hi,
Please find attached document for response.
Thank you.
